# Targeted and whole-genome sequencing reveal a north-south divide in *P. falciparum* drug resistance markers and genetic structure in Mozambique

Clemente da Silva[1,17], Simone Boene [1,17], Debayan Datta[2], Eduard Rovira-Vallbona[2], Andrés Aranda-Díaz[3,4], Pau Cisteró [2], Nicholas Hathaway[5], Sofonias Tessema[3], Arlindo Chidimatembue[1], Glória Matambisso[1], Abel Nhama [1,6], Eusebio Macete[1], Arnau Pujol [2], Lidia Nhamussua[1], Beatriz Galatas[1,2], Caterina Guinovart [2], Sónia Enosse[6], Eva De Carvalho[7], Eric Rogier [8], Mateusz M. Plucinski[9], James Colborn[10], Rose Zulliger[11], Abuchahama Saifodine[12], Pedro L. Alonso[1,13], Baltazar Candrinho[14], Bryan Greenhouse [3], Pedro Aide [1,6], Francisco Saute[1] & Alfredo Mayor [1,2,15,16✉]

Mozambique is one of the four African countries which account for over half of all malaria deaths worldwide, yet little is known about the parasite genetic structure in that country. We performed *P. falciparum* amplicon and whole genome sequencing on 2251 malaria-infected blood samples collected in 2015 and 2018 in seven provinces of Mozambique to genotype antimalarial resistance markers and interrogate parasite population structure using genome-wide microhaplotypes. Here we show that the only resistance-associated markers observed at frequencies above 5% were *pfmdr1*-184F (59%), *pfdhfr*-51I/59 R/108 N (99%) and *pfdhps*-437G/540E (89%). The frequency of *pfdhfr/pfdhps* quintuple mutants associated with sulfadoxine-pyrimethamine resistance increased from 80% in 2015 to 89% in 2018 (p < 0.001), with a lower expected heterozygosity and higher relatedness of microhaplotypes surrounding *pfdhps* mutants than wild-type parasites suggestive of recent selection. *pfdhfr/pfdhps* quintuple mutants also increased from 72% in the north to 95% in the south (2018; p < 0.001). This resistance gradient was accompanied by a concentration of mutations at *pfdhps*-436 (17%) in the north, a south-to-north increase in the genetic complexity of *P. falciparum* infections (p = 0.001) and a microhaplotype signature of regional differentiation. The parasite population structure identified here offers insights to guide antimalarial interventions and epidemiological surveys.

[1] Centro de Investigação em Saúde de Manhiça (CISM), Maputo, Mozambique. [2] ISGlobal, Hospital Clínic – Universitat de Barcelona, Barcelona, Spain. [3] EPPIcenter Research Program, Division of HIV, ID, and Global Medicine, Department of Medicine, University of California, San Francisco, CA, USA. [4] Chan Zuckerberg Biohub, San Francisco, CA, USA. [5] University of Massachusetts Chan Medical School, Worcester, MA, USA. [6] Instituto Nacional de Saúde (INS), Ministério da Saúde, Maputo, Mozambique. [7] World Health Organization, WHO Country Office Maputo, Maputo, Mozambique. [8] Malaria Branch, Division of Parasitic Diseases and Malaria, United States Centers for Disease Control and Prevention, Atlanta, GA, USA. [9] United States President's Malaria Initiative, Malaria Branch, Division of Parasitic Diseases and Malaria, United States Centers for Disease Control and Prevention, Atlanta, GA, USA. [10] Clinton Health Access Initiative, Maputo, Mozambique. [11] U.S. President's Malaria Initiative, USAID, Washington, DC, USA. [12] U.S. President's Malaria Initiative, USAID, Maputo, Mozambique. [13] Hospital Clinic-Universitat de Barcelona, Barcelona, Spain. [14] National Malaria Control Programme, Ministry of Health, Maputo, Mozambique. [15] Spanish Consortium for Research in Epidemiology and Public Health (CIBERESP), Madrid, Spain. [16] Department of Physiologic Sciences, Faculty of Medicine, Universidade Eduardo Mondlane, Maputo, Mozambique. [17] These authors contributed equally: Clemente da Silva, Simone Boene. ✉email: alfredo.mayor@isglobal.org

Mozambique is among the ten countries with the highest burden of malaria worldwide, with an estimated 10.2 million cases in 2021[1]. Malaria transmission is very heterogeneous in the country, with a high burden in the north and very low transmission in the south, therefore requiring different strategies for effective control and potential elimination[2]. Early treatment of malaria illness with artemisinin-based combination therapies (ACTs) and the use of antimalarial medicines for prophylaxis and prevention remain key to malaria control and, ultimately, malaria elimination. However, resistance to artemisinin[3] and partner drugs[4], as well as to sulfadoxine-pyrimethamine (SP) used for chemoprevention[5], threatens the global effort to reduce the burden of malaria[6].

Surveillance of antimalarial efficacy is key to mitigate and manage the risk of resistance to antimalarial drugs[4]. The identification of molecular markers of antimalarial resistance has led to genetic approaches that can complement therapeutic efficacy studies which follow standardized protocols[6,7] to confirm resistance, monitor trends and raise early warning signals[6]. In the case of artemisinin, partial resistance (delayed parasite clearance) has been linked to mutations in the *pfkelch13* propeller region[3,6]. In the Greater Mekong Subregion, emergence of these mutations has been associated with mutations in *P. falciparum* apicoplast ribosomal protein 10 (*pfarps10*; PF3D7_1460900), ferrodoxin (*pffd*, PF3D7_1318100), chloroquine resistance transporter (*pfcrt*; PF3D7_0709000), and multidrug resistance 2 (*pfmdr2*; PF3D7_1447900) genes[8]. Recently, the validated *pfkelch13* mutation R561H has been detected in Rwanda[9] and Tanzania[10], whereas A675V and C469Y have been associated with prolonged parasite clearance half-lives in Uganda[11].

The development of resistance to ACT partner drugs continues to pose a challenge in the treatment of malaria[4]. Increased resistance to piperaquine has been associated with a gene amplification of a section of chromosome 14 involving the genes *plasmepsin* 2 and 3[12], as well as with single nucleotide polymorphisms in a putative exonuclease gene (*pfexo*, PF3D7_1362500) in parasite isolates from Cambodia[12]. Mutations in the multidrug resistance transporter 1 (*pfmdr1*) gene (N86Y, Y184F, and D1246Y) have been associated but not fully validated with susceptibility to multiple drugs[4,6], including artesunate-amodiaquine and artemether-lumefantrine[13]. The K76T mutation at *pfcrt*, together with different sets of mutations at other codons (including C72S, M74I, N75E, A220S, Q271E, N326S, I356T, and R371I) has been linked to chloroquine resistance[4,6,14]. Finally, clinical treatment failure with SP has been linked to A437G and K540E mutations of dihydropteroate synthase (*pfdhps*) in combination with triple mutations (N51I + C59R + S108N) in dihydrofolate reductase (*pfdhfr*)[15]. Additional *pfdhps* mutations (S436A/C/F/H and A581G) have been suggested to increase the levels of SP resistance[16].

Identifying mutations associated with drug resistance from samples collected on a routine basis can inform drug policies and ensure that interventions utilize appropriate drug regimens. Since replacing chloroquine with a combination of amodiaquine and SP for uncomplicated malaria treatment in 2003, the Mozambique national treatment guidelines underwent various revisions[17]. In 2006, ACT was formally introduced by adopting artesunate/SP as a first-line treatment for uncomplicated *P. falciparum* malaria. The most recent change occurred in 2009, when the country introduced artemether-lumefantrine as the official first-line treatment, with artesunate-amodiaquine as a backup in situations when artemether-lumefantrine is contraindicated. Intermittent preventive treatment in pregnancy (IPTp) with SP was first implemented in the country in 2006, and delivered free of charge to all pregnant women[18]. In 2014, the national guidelines were updated and implemented countrywide to adjust to the ≥3 SP-dose World Health Organization recommendation. In 2015, a national household survey reported an IPTp-SP country

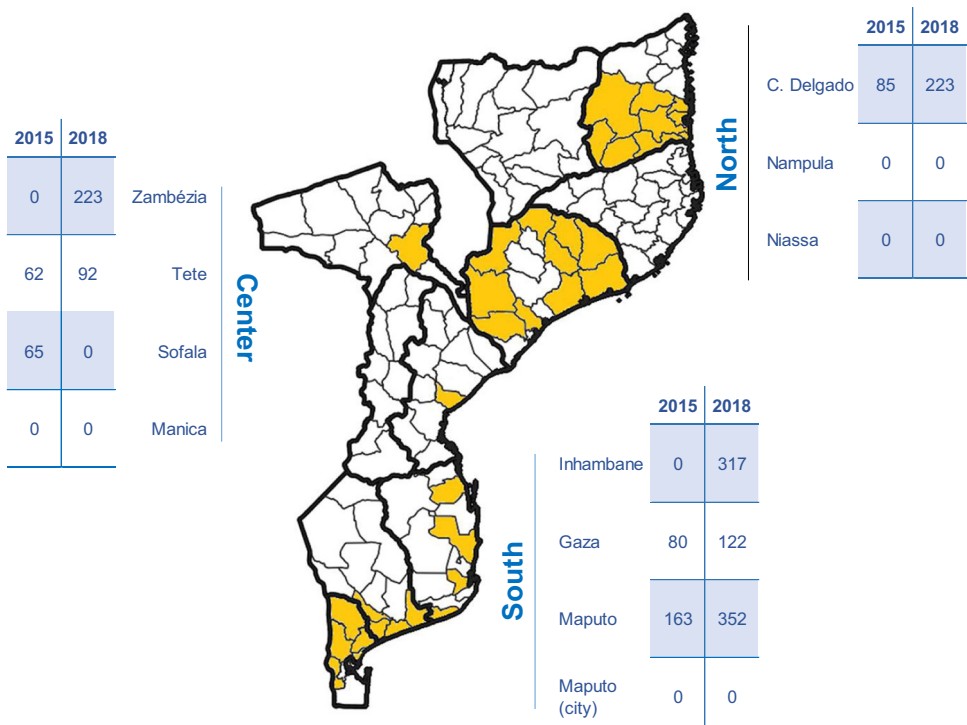

**Fig. 1 Source of *P. falciparum* samples providing genetic data.** Tables indicate the number of samples included in the analysis per province and year for each of the three main regions of the country. Provincial borders are indicated with thick lines. The specific districts providing data for the study are colored. Made with QGIS.

**Center**

| 2015 | 2018 | |
|---|---|---|
| 0 | 223 | Zambézia |
| 62 | 92 | Tete |
| 65 | 0 | Sofala |
| 0 | 0 | Manica |

**North**

| | 2015 | 2018 |
|---|---|---|
| C. Delgado | 85 | 223 |
| Nampula | 0 | 0 |
| Niassa | 0 | 0 |

**South**

| | 2015 | 2018 |
|---|---|---|
| Inhambane | 0 | 317 |
| Gaza | 80 | 122 |
| Maputo | 163 | 352 |
| Maputo (city) | 0 | 0 |

coverage of 51.4% for one dose, 34.2% for two doses, and 22.4% for ≥3 doses[19]. Currently, the country is piloting the use of seasonal (SP and amodiaquine) and perennial (SP) malaria chemoprophylaxis. Several studies have reported the prevalence of molecular markers of antimalarial resistance in Mozambique[14,20–23], but there is no comprehensive analysis of their spatial and temporal distribution in the context of the overall parasite genetic structure. In this study, we used amplicon-based and whole genome sequencing, machine-learning approaches, and relatedness as well as diversity analysis of micro-haplotypes flanking *pfdhps* to describe the spatial and temporal distribution of antimalarial drug resistance markers, the geographic structure of *P. falciparum* parasites, and the evolutionary history of *pfdhps* mutant alleles in samples collected in 2015 and 2018 across south, central and north Mozambique.

## Results

**Sample size and geographic distribution.** Among the 2251 *P. falciparum* samples included in this study, sequencing produced at least one resistance-associated genotype (among 11 genetic markers targeted) in 1784 (79%) samples (455 from 2015 and 1329 from 2018; 308 from North, 440 from Central, and 1034 from South Mozambique; Fig. 1 and Supplementary Tables 1–3). Among these samples, 1522 were obtained from malaria clinical cases (therapeutic efficacy studies, health facility surveys, or reactive surveillance), 200 from community surveys (mass drug administration, cross-sectional surveys), and 62 from pregnant women at first antenatal care visits (Supplementary Table 1). Whole genome sequences were obtained from a total of 1452 (64%) samples which passed quality filters.

**Polymorphisms in *pfkelch13* gene and artemisinin-resistance predisposing background.** Among the 1429 *P. falciparum* samples successfully genotyped for *pfkelch13*, 1393 were fully wild-type and 36 (2.5%) presented a total of 32 non-synonymous mutations not associated with artemisinin tolerance (Table 1). A mutation in codon 537 (N537D) was observed in a sample from southern Mozambique (2018). Of the six amino acids making the artemisinin-resistance genetic background, only *pfcrt* N326Y showed any variation, with five isolates out of 1637 (0.3%) carrying a mixed genotype (Table 2). Similarly, no mutations were observed at codon 415 of *pfexo* associated with resistance to piperaquine (*n* = 1394). The *plasmepsin2/3* breakpoint was detected in 2 (0.4%) out of 524 *P. falciparum* isolates (Table 2).

**Polymorphisms in *pfcrt* and *pfmdr1*.** Mutations at codons 72 (*n* = 1655), 74 (*n* = 1657), 75 (*n* = 1658), 76 (*n* = 1656) in *pfcrt*, and at codons 86 (*n* = 1605), and 1246 in *pfmdr1* (*n* = 1519) were absent or below 5% (Table 2). In contrast, 59% (899/1536) of the samples tested carried mutations at codon 184 (534 pure mutants and 365 mixed genotypes; Supplementary Tables 4, 5). No statistically significant difference was observed in the carriage of this mutation between provinces or study periods (Supplementary Fig. 1 and Supplementary Tables 6–8).

**Polymorphisms in *pfdhfr* and *pfdhps* genes.** Mutations at codons 164 in *pfdhfr*, and 581 and 613 in *pfdhps* were either absent or below 1% (Table 2). Mixed genotypes were observed at frequencies of 1–2% for 108, 51, and 59 *pfdhfr* codons, and 5–11% for 437 and 540 *pfdhps* codons (Supplementary Table 5). After excluding these mixed genotypes, the overall frequency of mutations in *pfdhfr* was ≥97% (97% in codon 51 [1596/1638], 98% in codon 59 [1597/1625] and 99% in codon 108 [1635/1649]) and ≥88% in *pfdhps* (90% in codon 437 [1289/1439] and 88% in codon 540 [1242/1404]; Supplementary Table 6 and

Supplementary Fig. 2). The most prevalent *pfdhfr* and *pfdhps* alleles were the triple (S108N/N51I/C59R; 99% [1548/1600]) and double mutants (A437G/K540E; 89% [1228/1377]), respectively, with an 87% (1155/1330) of quintuple mutants (Supplementary Table 6). The overall frequency of quintuple mutants increased from 80% [234/293] in 2015 to 89% [921/1037] in 2018 (*p* < 0.001; Fig. 2a–c, Supplementary Table 7 and Supplementary Data 1), mainly in Cabo Delgado (from 40 to 72%, *p* < 0.001) and Gaza (from 90 to 100%, *p* < 0.001). Similar increases were observed for triple *pfdhfr* and double *pfdhps* mutants (*p* < 0.001). The frequency of quintuple mutants increased from north to south, both in 2015 (40% in Cabo Delgado vs 93% in Maputo; *p* < 0.001) and 2018 (72% in Cabo Delgado vs 95% in Maputo; *p* < 0.001), mainly driven by differences in *pfdhps* double mutants (Fig. 2a–c). The multivariable logistic regression analysis showed that both region (north, central and south) and period (2015 and 2018) were independently associated with the relative abundance of *pfdhfr/dhps* mutations, which increased from north to south and from 2015 to 2018 (Supplementary Table 8).

The distribution of mutations at codon 436 (S436C/A/H/F) in *pfdhps* had a very marked geographic pattern (Fig. 2d and Supplementary Data 1). After excluding mixed genotypes, mutations at 436 were observed in 17% (40/232; C in 6, F in 5, H in 4, and A in 25) of the isolates obtained from Cabo Delgado, but only in 0.6% (8/1307) of the isolates from the rest of the country, and never in combination with a double 437/540 mutation background (Supplementary Table 9). Therefore, three different *pfdhps* haplotypes were observed in Cabo Delgado: triple

| **Table 1 *Pfkelch13* mutations detected in *P. falciparum* isolates collected in 2015 and 2018 in seven provinces from Mozambique.** | | | |
|---|---|---|---|
| **Mutation** | **n** | **Province** | **Year** |
| K372E | 1 | Maputo | 2015 |
| H384R | 1 | Maputo | 2018 |
| V386I | 1 | Sofala | 2015 |
| V386L | 1 | Maputo | 2018 |
| R404K | 1 | C. Delgado | 2015 |
| G436S | 1 | Gaza | 2015 |
| **F442L** | 1 | Maputo | 2018 |
| **V454I** | 2 | C. Delgado | 2015 |
| **K480R** | 1 | Gaza | 2015 |
| F483S | 1 | Gaza | 2015 |
| F483L | 1 | Maputo | 2018 |
| **S485G** | 1 | C. Delgado | 2015 |
| **V494I** | 1 | Maputo | 2018 |
| V520F | 1 | Maputo | 2018 |
| **N537D** | 1 | Maputo | 2018 |
| **G544R** | 1 | C. Delgado | 2015 |
| Y546H | 1 | Sofala | 2015 |
| P553T | 1 | Gaza | 2015 |
| **A578S** | 4 | Tete/C. Delgado | 2015/2018 |
| **E596G** | 1 | Zambezia | 2018 |
| R597G | 1 | C. Delgado | 2015 |
| L598S + N599S | 1 | Maputo | 2015 |
| N599D | 1 | C. Delgado | 2018 |
| **E605G** | 1 | Maputo | 2015 |
| **K607E** | 1 | Maputo | 2018 |
| L631F | 1 | Maputo | 2018 |
| **D641N** | 1 | Maputo | 2015 |
| **F656I** | 1 | Sofala | 2015 |
| R659G | 1 | Maputo | 2015 |
| **Q661R** | 1 | C. Delgado | 2015 |
| F662S | 1 | C. Delgado | 2015 |

In bold, those mutations reported in previous studies[24, 64, 70, 71].

**Table 2 Molecular markers of *P. falciparum* antimalarial resistance observed at frequencies below 5% in Mozambique.**

| | | North | | Center | | South | |
|---|---|---|---|---|---|---|---|
| | | *n* | % | *n* | % | *n* | % |
| *pfcrt* | | | | | | | |
| C72S | wt | 302 | 100 | 424 | 100 | 928 | 100 |
| M74I | wt | 302 | 100 | 425 | 100 | 918 | 98.7 |
| | mut | 0 | 0 | 0 | 0 | 10 | 1.1 |
| | mix | 0 | 0 | 0 | 0 | 2 | 0.2 |
| N75E | wt | 302 | 100 | 425 | 100 | 918 | 98.6 |
| | mut | 0 | 0 | 0 | 0 | 10 | 1.1 |
| | mix | 0 | 0 | 0 | 0 | 3 | 0.3 |
| K76T | wt | 302 | 100 | 425 | 100 | 917 | 98.7 |
| | mut | 0 | 0 | 0 | 0 | 10 | 1.1 |
| | mix | 0 | 0 | 0 | 0 | 2 | 0.2 |
| *Artemisinin-resistance genetic background* | | | | | | | |
| V127M (*pfarps10*) | wt | 304 | 100 | 437 | 100 | 965 | 100 |
| D128Y/H (*pfarps10*) | wt | 225 | 100 | 386 | 100 | 596 | 100 |
| D193Y (*pffd*) | wt | 298 | 100 | 415 | 100 | 901 | 100 |
| N326S (*pfcrt*) | wt | 292 | 99.3 | 413 | 99.8 | 932 | 99.8 |
| | mix | 2 | 0.7 | 1 | 0.2 | 2 | 0.2 |
| I356T (*pfcrt*) | wt | 297 | 100 | 419 | 100 | 924 | 100 |
| T484I (*pfmdr2*) | wt | 297 | 100 | 412 | 100 | 856 | 100 |
| *Plasmepsin 2/3 breakpoint* | | | | | | | |
| | wt | 117 | 99.2 | 180 | 100 | 225 | 99.6 |
| | Duplication | 1 | 0.8 | 0 | 0 | 1 | 0.4 |
| *pfexo* | | | | | | | |
| E415G | wt | 241 | 100 | 300 | 100 | 853 | 100 |
| *pfmdr1* | | | | | | | |
| N86Y | wt | 290 | 99.7 | 416 | 100 | 888 | 98.9 |
| | mut | 1 | 0.3 | 0 | 0 | 7 | 0.8 |
| | mix | 0 | 0.0 | 0 | 0 | 3 | 0.3 |
| D1246Y | wt | 285 | 100 | 396 | 100 | 837 | 99.9 |
| | mix | 0 | 0 | 0 | 0 | 1 | 0.12 |
| *pfdhfr* | | | | | | | |
| I164L | wt | 294 | 100 | 412 | 99.8 | 864 | 100 |
| | mix | 0 | 0 | 1 | 0.24 | 0 | 0 |
| *pfdhps* | | | | | | | |
| A581G | wt | 287 | 99.7 | 376 | 99.5 | 824 | 100 |
| | mut | 0 | 0.0 | 1 | 0.3 | 0 | 0 |
| | mix | 1 | 0.3 | 1 | 0.3 | 0 | 0 |
| A613S/T | wt | 281 | 98.3 | 375 | 99.5 | 827 | 100 |
| | mut | 1 | 0.3 | 0 | 0.0 | 0 | 0 |
| | mix | 4 | 1.4 | 2 | 0.5 | 0 | 0 |

*wt* wild-type, *mut* mutant, *mix* mixed genotypes

wild-type (S436/A437/K540; 26/203 [13%]), mutant at codon 436 but wild-type in codons 437 and 540 (37/203 [18%]), and wild-type at codon 436 but mutant in codons 437 and 540 (S436/A437G/K540E; 139/203 [68%]). In contrast, the S436/A437G/K540E haplotype was predominant in the rest of the country (1089/1162 [94%]; Supplementary Table 9). No change in the frequency of mutations in codon 436 was observed between study periods ($p = 0.371$; Supplementary Table 8).

**Population structure.** A total of 8722 microhaplotype loci were reconstructed via local assembly from 1438 samples which produced whole genome sequences. Of these, 349 samples contained data for less than 50 percent of all microhaplotype loci (Supplementary Fig. 3a) and were therefore excluded. The median expected heterozygosity ($H_e$) of the 8722 microhaplotypes from the 1089 samples with data for more than 50% of the microhaplotypes was 0.312 (Interquartile range [IQR]: 0.196–0.498). Twenty-four percent of the microhaplotypes had high expected heterozygosity ($H_e > 0.5$) in the parasite population analyzed, and 366 had $H_e > 0.75$ (Fig. 3a and Supplementary Data 1).

The 25% most diverse microhaplotype loci ($n = 2181$) were evaluated as predictors for geographic classification using a random forest analysis at the province and regional levels. The model failed to classify samples at the province level (Out-of-bag error rate = 50.51%). However, the out-of-bag error rate was

24.89% at the regional level (North-Central-South; Fig. 3c, d and Supplementary Table 10). The lowest out-of-bag error rate was observed when classifying samples from North and South (8%), and higher rates when central region samples were considered (15.26% for Central-South and 36.79% for North-Central; Supplementary Fig. 4). Removal of 155 microhaplotypes caused the model to lose accuracy in prediction to the regional classification below the inflexion point of the distribution of mean decrease in accuracy (Supplementary Fig. 3b), and were therefore considered as the most relevant. Thirty-one percent of these microhaplotypes were located in chromosome 6, followed by percentages below 10% in the rest of the chromosomes (Fig. 3b and Supplementary Data 1).

Overall within-host complexity of *P. falciparum* infections, calculated from the 100 microhaplotypes with the highest $H_e$, was 2 (IQR [1,2]) with a prevalence of 47% (517/1090) monogenomic infections. The complexity of infection (COI) and prevalence of monoclonal infections in 2015 was similar in the three regions from Mozambique ($p = 0.801$ and $p = 0.507$, respectively). However, median COI in 2018 differed between the three regions ($p < 0.001$), with the lowest values in the south (1, IQR [1,2]), followed by the center (2, IQR [1,2]) and north (2, IQR [1,3]). Similar trends were observed in the prevalence of monogenomic infections (51% in the south, 46% in the center, and 35% in the north; $p = 0.005$; Fig. 3e, f, Supplementary Tables 8, 11, and Supplementary Data 1).

Microhaplotypes flanking *pfdhps* were used to infer the evolutionary history of the mutant alleles in Cabo Delgado, as in the rest of the provinces, double mutants had almost reached fixation (frequencies between 80 and 100%). Sixteen microhaplotypes were contained in a 50 kb region around the gene *pfdhps*, 15 of them in eight genes and one intergenic (Supplementary Fig. 5). These flanking microhaplotypes separated the parasites carrying the double *pfdhps* mutant haplotype (always accompanied by a wild-type 436 codon) from the rest of parasites (Fig. 4a, b and Supplementary Data 1). The 50 kb region flanking *pfdhps* was more similar among double *pfdhps* mutants ($n = 92$; median identity by state [IBS] = 0.88, IQR[0.81–0.91]) than among the double wild-type ($n = 51$, median IBS = 0.68, IQR[0.62–0.76]; $p < 0.001$; Supplementary Fig. 6). Similarly, $H_e$ of microhaplotypes flanking *pfdhps* was 60% lower in the double mutants (median = 0.1, IQR[0.04–0.26]) than in wild-type alleles (median = 0.34, IQR[0.21–0.41]; $p = 0.016$; Fig. 4c, Supplementary Table 12, and Supplementary Data 1), consistent with recent selection.

## Discussion

This study provides a country-wide resolution of *P. falciparum* markers of antimalarial resistance and genetic structure in Mozambique which can be used to inform the use of antimalarials for treatment and chemoprevention as well as to study the impact of future interventions. The genomic data provides evidence that: (1) although non-synonymous mutations were observed in *pfkelch13*, none of them have been associated with artemisinin tolerance; (2) genetic variants associated with piperaquine and chloroquine resistance were rare in 2018; (2) in contrast, the frequency *pfdhfr/dhps* mutations increased from north to south, almost reaching fixation in Maputo Province; and (3) this spatial trend was accompanied by a reduction towards the south in the genetic complexity of *P. falciparum* infections and a signal of geographic differentiation which allows a regional separation based on highly diverse microhaplotypes.

The *pfkelch13* wild-type, artemisinin-sensitive haplotype predominated in the parasite population surveyed from Mozambique, and none of the artemisinin-resistant validated variants were

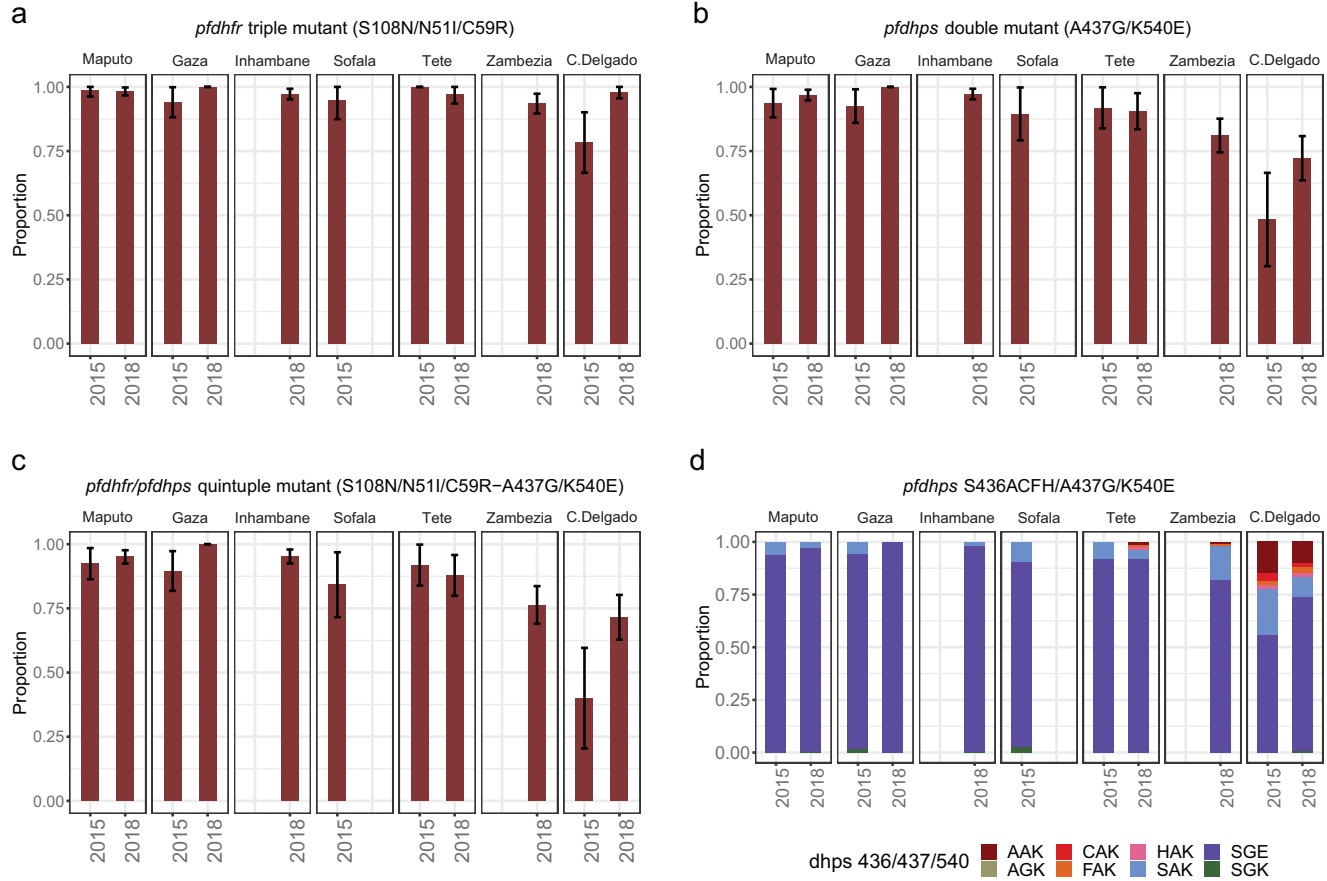

**Fig. 2 Molecular markers of *P. falciparum* sulfadoxine-pyrimethamine (SP) resistance in Mozambique.** Frequency of *P. falciparum* isolates carrying triple mutations in *pfdhfr* (**a**), double mutations in *pfdhps* (**b**), and quintuple mutations in *pfdhfr/phdhps* (**c**) in 2015 and 2018 in seven provinces from Mozambique. For the *pfdhps* haplotype 436/437/540 (**d**), frequencies of the different allelic combinations are shown ($n = 1365$). Frequencies were calculated after excluding mixed genotypes. Data from Sofala was only available for 2015, and from Inhambane and Zambézia for 2018. The error bars represent a 95% confidence interval for the population proportion.

detected[6]. However, an array of 32 rare non-synonymous mutations were identified, similar to results from other studies in Africa[24]. Among them, the N537D mutation, which has been reported as potentially associated with delayed clearance[24], was detected in one sample, whereas A578S, the most predominant *pfkelch13* mutation in *P. falciparum* parasites of African origin[24] but not associated with artemisinin tolerance, was found in four of the 1429 genotyped samples. The background mutations found to anticipate the emergence of *pfkelch13* mutations in South-East Asia[8] were not detected in this study. Similarly, there is no strong evidence of polymorphisms associated with resistance to ACT partner drugs. Only 0.4% of the samples analyzed showed evidence of piperaquine resistance, based on the analysis of the breakpoint within the distal end of *plasmepsin 3*[25] associated with *plasmepsin 2/3* duplications and the single nucleotide polymorphism at codon 415 of the putative exonuclease gene[12]. This is in agreement with a previous study in Mozambique which found multiple copies of *plasmepsin 2* in 1.1% of the samples analyzed[21]. Mutations in codon 86 of *pfmdr1*, associated with resistance to amodiaquine and increased susceptibility lumefantrine[26,27], were detected in 11 of the 1605 samples analyzed. Fifty-nine percent of the parasites carried the 184 F *pfmdr1* variant, although this mutation appears to have a weaker association with antimalarial effectiveness in vivo[28,29] and in vitro[26]. However, *pfmdr1* markers must be considered with caution, due to inconsistent associations with ACT partner drug resistance[30], pointing out that robust

molecular markers associated with amodiaquine and lumefantrine are still lacking.

The present study revealed an evolutionary process acting on the molecular markers of SP resistance. Overall, a high frequency of triple *pfdhfr* (99%), double *pfdhps* (89%), and quintuple mutant haplotypes (87%) was observed, which increased from 2015 to 2018 and from north to south. Microhaplotypes in the 50 kb region around *pfdhps* mutant alleles were more similar (higher IBS) and less diverse (lower expected heterozygosity) than around the wild-type allele, suggesting a recent expansion of the double mutant population in the country[31]. Geographical heterogeneities in the prevalence of *pfdhfr/dhps* aleles were accompanied by a different distribution of *pfdhps*-436 mutation, which was only detected in the north of the country (Cabo Delgado) at a frequency of 17% and never in combination with the double 437 and 540 mutant haplotypes. Changes at codon 436 have been associated with higher levels of in vitro SP resistance[32], although in vivo resistance evidences are less clear[33]. The increase in *pfdhps* mutations from north to south was also accompanied by a reduction in the number of genetically distinct parasite strains infecting an individual, indicative of declining malaria transmission intensity[34]. Finally, the *pfdhps* mutational pattern was also coincident with a regional separation of the parasite population based on highly diverse microhaplotypes, suggestive of geographical structuring. Geographical distance, barriers in gene flow across the regions, and differences in the coverage of antimalarial interventions[35] due to unequal

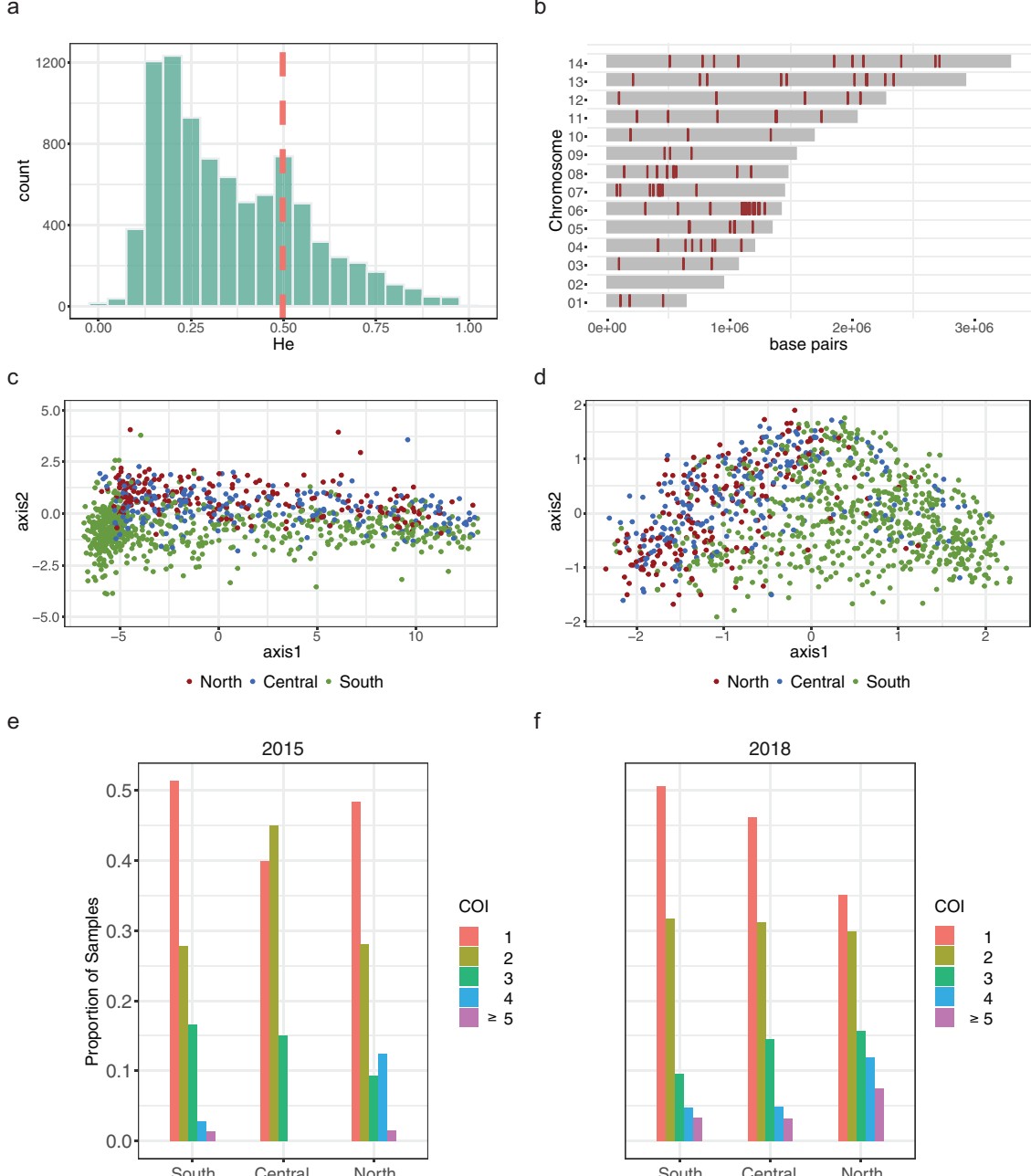

**Fig. 3 P. falciparum population structure by geography in Mozambique.** Microhaplotypes from regions of 150–300 bp in length between long tandem repeats were reconstructed from whole genome sequences and used to test the geographic structure of *P. falciparum* parasites. **a** Distribution of the expected heterozygosity at the 8722 microhaplotype loci extracted from whole genome sequences. The y-axis represents the number of microhaplotype loci for a given expected heterozygosity. The red line marks the 75% percentile of the distribution; the 25% most diverse loci were considered for population structure analysis. **b** Chromosomal locations of the 155 most important microhaplotypes, which contribute to the geographic (North-Central-South) classification model. **c** Principal coordinates analysis with samples grouped into regions (North-Central-South; $n = 1089$), considering microhaplotypes at loci with expected heterozygosity in the top 25% percentile. **d** Principal coordinates analysis with samples grouped into regions considering the 155 top microhaplotypes, with an out-of-bag error rate of classification of 24.89%. **e, f** Complexity of infection (COI) for samples in different regions of Mozambique in 2015 (**e**) and 2018 (**f**), as indicated by the number of genetically distinct clones. Regional assignment of samples: North: C. Delgado; Central: Sofala, Tete, and Zambézia; South: Gaza, Inhambane, and Maputo.

distribution of resources and security issues, could have contributed to the microhaplotype regional differentiation, which might have affected the geographic patterns observed in the molecular markers of SP resistance. However, it is still unclear what selective forces have fueled the spread of *pfdhfr/dhps* mutants in the absence of its large-scale use, as Mozambique abandoned SP for clinical management in 2009. Compensatory

mechanisms that reduce the mutation fitness cost[36] or an insufficient pool of sensitive parasites to fuel recovery[37], may have contributed to the increase in *pfdhfr/pfdhps* mutants in the absence of SP drug pressure. However, these were not limiting factors for the recovery of chloroquine sensitivity in Mozambique where the mutations in *pfcrt* were almost fixated[38]. The contribution to drug pressure of SP for IPTp or other drugs such

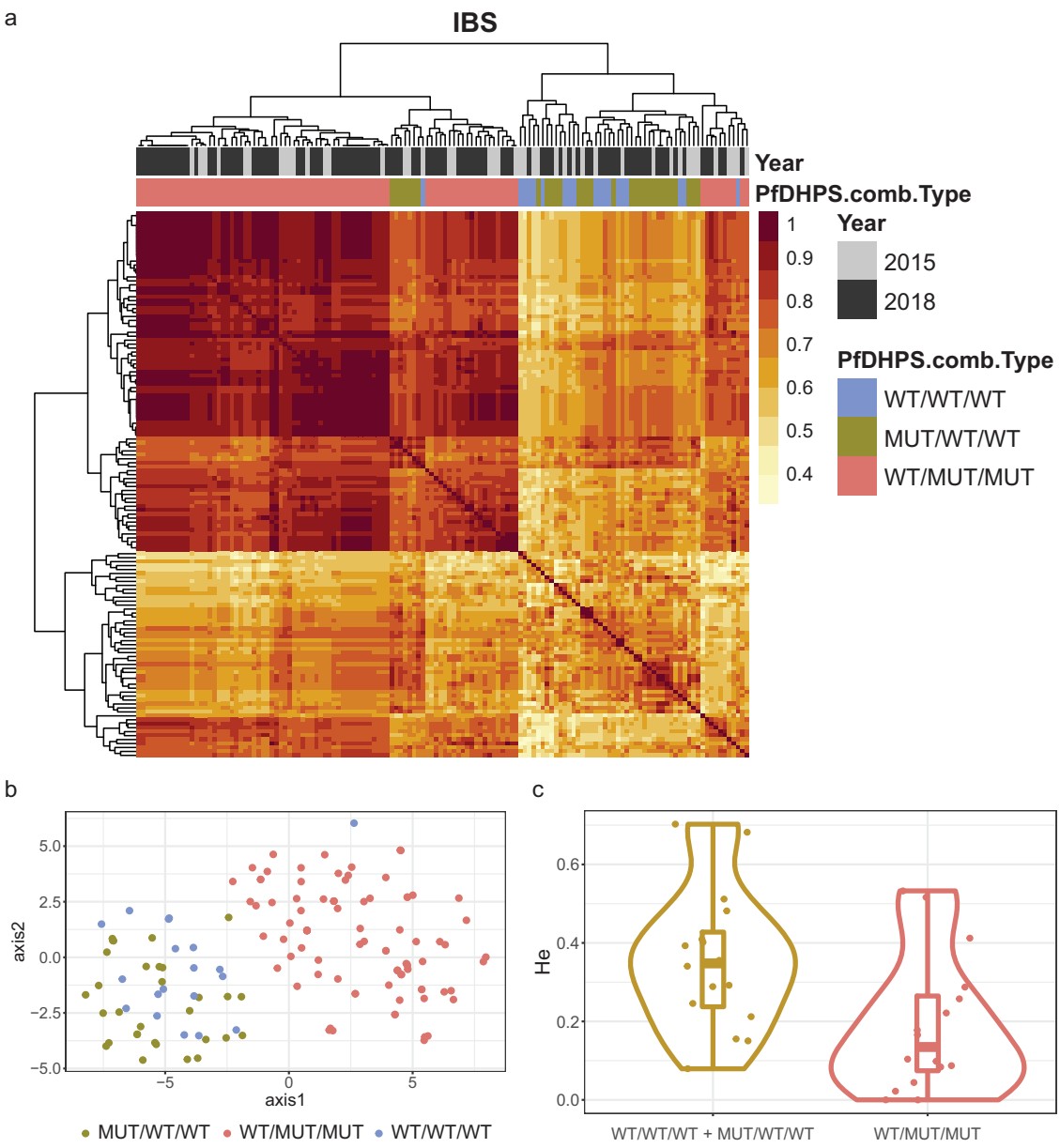

**Fig. 4 Regional separation, relatedness, and expected heterozygosity of *pfdhps* allelic haplotypes.** Identify by state (IBS) and expected heterozygosity ($H_e$) was calculated using 16 microhaplotypes flanking *pfdhps* to assess the evolutionary history of *pfdhps* mutant alleles in Mozambique. **a** Heatmap of the inter-sub-population IBS matrix among *dhps* alleles in Cabo Delgado observed in 2015 and 2018 (wild-type in codons 436, 437, and 540 [WT/WT/WT]: $n = 20$; mutant in codon 436 but wild-type in codons 437 and 540 [MUT/WT/WT]: $n = 31$; wild-type in codon 436 but mutant in codons 437 and 540 [WT/MUT/MUT]: $n = 92$). Sixteen microhaplotypes in a 50 kb region around *pfdhps* were used to calculate the pairwise IBS between samples. **b** t-distributed stochastic neighbor embedding visualization after 10000 iterations and **c** Expected heterozygosity calculated from the 16 microhaplotype loci in a 50 kb region around *pfdhps* in parasites collected from Cabo Delgado. Median and interquartile (IQR) $H_e$ values: 0.1, IQR (0.04–0.26) for double mutants; 0.37, IQR (0.2–0.47) for WT/WT/WT; and 0.28, IQR (0.13–0.4) for MUT/WT/WT. The lower, middle, and upper hinges of the rectangle correspond to the 25% quantile, median, and 75% quantile of the distribution, respectively.

as cotrimoxazole[39,40] is likely to be small, as the populations targeted make up only a small part of the overall population of Mozambique at a given time. Finally, lower levels of antimalarial immunity and sexual recombination to unlink resistance haplotypes may have also contributed to the higher carriage of molecular markers of SP resistance in southern Mozambique, where malaria transmission is the lowest[41]. Although these patterns seem consistent with directional selection due to drug pressure, the nature of the study does not allow to discard other factors, such as the regional impact of neighboring countries' drug policies[42].

The results of this study have several public health implications. First, all available in vivo efficacy data[7,43,44] and the lack of validated *kelch13* mutations in 2018 suggest the appropriate efficacy of artemisinin for *P. falciparum* treatment and reduction of malaria transmission in Mozambique. However, the broad array of rare non-synonymous mutations that were detected could potentially provide a deep reservoir of variations for the emergence of artemisinin tolerance[45], as has been recently reported in Rwanda and Uganda[9,11]. Second, the lack of mutations at codon 415 of *pfexo* and the low prevalence of *plasmepsin2/3* breakpoints detected (0.4%) suggests the appropriate

performance of piperaquine as an ACT partners drug. However, *plasmpesin2/3* gene amplification should be closely monitored, given the rapid emergence and spread of piperaquine resistance in Southeast Asia, resulting in high treatment failure rates after dihydroartemisinin-piperaquine treatment[12]. Third, the data reassures the use of SP for chemoprevention in spite of the high carriage of *pfdhps and pfdhfr* quintuple mutants, due to the lack of evidence of a relationship between molecular markers and chemopreventive efficacy[5,46]. Moreover, the *pfdhps* A581G mutation, which has been suggested to reduce the SP chemopreventive effectiveness in infants and pregnant women[47–49], was detected only in three of the 1490 (0.2%) samples that were analyzed, therefore supporting the continued use of SP for IPTp in Mozambique. Similarly, amodiaquine effectiveness is likely to remain acceptably high for seasonal malaria chemoprevention because of the very low prevalence of 72–76 mutations in *pfcrt* and the 86Y–184Y haplotype of *pfmdr1*, suggested to be necessary for clinically relevant resistance to amodiaquine in Africa[50,51]. Fourth, the very low prevalence (0.6%) of markers of chloroquine resistance in *pfcrt* and the evidence of a return of its therapeutic efficacy in Mozambique[38], together with its chemoprophylactic activity and safety profile, suggest that chloroquine could play a role, on its own or in combination with other drugs or tools, for chemoprevention at the population level or for currently unprotected populations such as first trimester pregnant women[52]. Fifth, microhaplotypes indicative of regional population structure may be useful for identifying broad-scale parasite flow or assignment of geographic origin in Mozambique. However, the lack of a finer population structure at the province level may impose some limitations on this approach, which might be better evaluated by taking advantage of differences in pairwise relatedness of parasites as opposed to regional differentiation in allele frequencies[53,54]. These highly diverse microhaplotyes may also be useful to develop parasite genetic diversity metrics for detecting changes in transmission intensity and monitor the effectiveness of antimalarial interventions[34]. Sixth, this study also shows the utility of secondary analysis of blood samples from other studies to describe molecular patterns of surveillance interest. And finally, heterogeneity in molecular markers of antimalarial resistance within Mozambique highlights the need to use caution when extrapolating survey results from a single location.

The study has several limitations. First, the dried blood spots that were evaluated represent a convenience sample obtained from different studies, resulting in heterogeneities in the age, the clinical status of individuals, and intensities of transmission that could affect the level of immunity and intake of treatment. While further work is required to quantify the impact of these factors, these heterogeneities represent those that can be found in a country such as Mozambique, where malaria transmission ranges from a very high burden in the north and very low in the south. Second, the exclusion of mixed wild-type-resistant infections to calculate the frequency of resistance haplotypes may bias resistance estimates[55]. Third, selective forces other than those driven by the use of antimalarials cannot be discarded, as the signals of recent selection were inferred using microhaplotypes in the 50 kb region around *pfdhps* which overlap coding regions of other genes. Finally, caution is required when inferring the treatment and chemopreventive efficacy of an antimalarial, which depend on factors other than intrinsic parasite susceptibility, such as patient-acquired immunity, initial parasite biomass, treatment adherence, dosing, drug quality, and pharmacokinetics[6]. However, information on molecular markers plays an important role in tracking resistance and should be leveraged to detect early warning signals[56]. Combining chemoprevention efficacy studies[57]

with the monitoring of *pfdhps* mutations is required to evaluate SP-based chemopreventive strategies.

In conclusion, this report shows north-south genetic signals of increasing molecular makers of SP resistance, decreasing genetic complexity of infections, and geographic differentiation in Mozambique. However, the very low prevalence of 581 mutations in *pfdhps* reassures the role of SP for chemoprevention in Mozambique. Similarly, no molecular signals of artemisinin tolerance were observed in Mozambique. These results provide baseline data for studying the evolution of *P. falciparum* parasites in response to changing national malaria treatment guidelines. Moreover, these findings prompt the integration of molecular surveillance systems with treatment and chemoprevention efficacy studies to track the emergence and expansion of drug resistance in Mozambique. To achieve this, addressing inefficiencies in sampling and sequencing efforts, together with financial support and appropriate use of the data generated, is required to ensure the sustainability of malaria molecular surveillance programs[58].

## Methods

**Study site and sample collection.** This study analyzed 2251 samples collected in 2015 ($n = 724$) and 2018 ($n = 1527$) from 40 districts in seven provinces from Mozambique (Supplementary Tables 1, 2): one in the north region (Cabo Delgado), three in the center region (Zambézia, Sofala, and Tete), and three in the south (Gaza, Inhambane, and Maputo; Fig. 1). Dried blood spots were collected from *P. falciparum*-infected individuals identified during six malaria observational studies and clinical trials conducted in 2015 and 2018[7,43,59–61]. In 2018, two health facility survey studies recruited individuals attending outpatient services in Maputo, Zambézia, Cabo Delgado, Inhambane, and Gaza (all ages)[59,60]. Samples from an additional two therapeutic efficacy studies included children less than 5 years of age with confirmed malaria (by rapid diagnostic tests) in Cabo Delgado, Tete, Sofala, and Gaza province in 2015[43] and Cabo Delgado, Tete, Zambézia, and Inhambane in 2018[7]. In the fifth study, all age individuals with a malaria-positive rapid diagnostic test were identified through community-based cross-sectional surveys in Maputo Province (2015 and 2018)[61], including a malaria elimination project area[61], which collected samples from individuals participating in mass drug administration campaigns and reactive surveillance in Magude District. Finally, in the sixth study, pregnant women at their first antenatal care visit with a *P. falciparum* infection confirmed by quantitative real-time PCR were identified through antenatal care surveys conducted in Maputo Province (2018)[62]. Health facility-based sampling sites were district or subdistrict health centers or provincial hospitals, selected by the Centro de Investigação em Saúde de Manhiça (CISM) or National Malaria Control Program according to their public health or research needs, whereas cross-sectional surveys were community-based and participants were randomly selected. Further information on sampling for each study is available in the associated publications. Before administering treatment, 50 μL dried blood spots on filter paper were obtained from each patient through fingerprick, identified with anonymous barcodes, and stored at 4 °C with silica gel.

**Inclusion and ethics.** Clinical-demographic data and blood samples were collected only after written informed consent and assent from all participants, or an accompanying adult, if younger than 18 years of age, was provided. All study protocols were approved by the Mozambican National Committee for Bioethics in Health. The research included local researchers throughout the research process, including the study design, study implementation, data ownership, intellectual property, and authorship of the publication. The research is locally relevant, as determined in collaboration with local partners, who agreed on the importance of malaria molecular surveillance. Roles and responsibilities were agreed amongst collaborators before implementing the research activities. Special emphasis has been allocated to capacity-building for local researchers on genomic and bioinformatic tools for molecular surveillance.

**Amplicon-based sequencing.** DNA was extracted from samples at the Malaria-GEN Laboratory at the Wellcome Sanger Institute, Hinxton, UK, using high-throughput robotic equipment (Qiagen QIAsymphony)[63]. Parasite DNA was amplified by applying selective whole genome amplification and genotyping was performed by the SpotMalaria platform[63]. Briefly, a first PCR was done to generate 190–250 bp amplicons of interest in the parasite genome using locus-specific multiplexed primers, followed by a second PCR to incorporate unique sample-level and primer-pool multiplexing adapters. After sequencing multiple samples on a single MiSeq lane, sequences were de-plexed using the unique multiplexing adapter IDs and aligned to a modified amplicon *P. falciparum* reference genome. Genotypes were called for each variant analyzed using bcftools and custom scripts[63],

namely: *pfkelch13* (any mutation in codons 349–726 corresponding to BTB/POZ and propeller domains)[64], *pfdhfr* (codons 51, 59, 108, 164)[15], *pfdhps* (codons 436, 437, 540, 581, 613)[16], *pfcrt* (codons 72, 73, 74, 75, 76)[4,6,14], *pfexo* (codon 415)[12], *pfmdr1* (codons 86, 184, 1246)[13], and artemisinin-resistance genetic background (codons 127, 128 in *pfarps10*, 193 in *pffd*, 326, 356 in *pfcrt* and 484 in *pfmdr2*)[8]. An assay designed to detect the breakpoint within the distal end of *plasmepsin 3* that includes the complete duplication of the *plasmepsin 2* gene (*plasmepsin2/3* breakpoint) was used to detect the hybrid sequence created as a result of the *plasmepsin 2/3* duplication[25].

**Whole genome sequencing**. *P. falciparum* samples were also whole genome sequenced at Wellcome Sanger Institute and the University of California, San Francisco. In brief, short sequence reads (200 bp) were generated on the Illumina HiSeqX platform at Wellcome Sanger Institute[65]. At the University of California, San Francisco, barcoded libraries prepared using the NEBNext Ultra II DNA Library Prep Kit after selective whole genome amplification[66] were pooled and sequenced on the Illumina NovaSeq 6000 System using 150 bp paired-end sequencing. Reads were filtered for a minimum per base quality of 20. Variant calls were generated by running a custom pileup program and filtered to have a minimum read depth of 10 and a minimum within-sample frequency of 5%, which were generated by utilizing selective whole genome amplification control runs on known lab strains to remove all false variant calls.

**Data analysis**. The analysis aimed at describing the spatial and temporal distribution of antimalarial drug resistance markers, the geographic structure of *P. falciparum* parasites and the evolutionary history of *pfdhps* mutant alleles. The frequency of infections carrying parasites with markers of antimalarial resistance were estimated at the province level, based on sampling location and year. For each codon, samples were classified as wild-type, mutant, or mixed if both wild-type and mutant alleles were detected or missing if samples failed to produce a valid genotype. Samples with no mixed genotypes were retained for the haplotype reconstruction and downstream statistical analysis. A local haplotype reconstruction tool (Pathweaver[67]) was used to extract microhaplotypes from regions of 150–300 bp in length between long tandem repeats. Microhaplotypes were selected to contain no homopolymers/dinucleotide repeats longer than 10 bp or length variation >3 bp, with at least two single nucleotide polymorphisms[60]. Samples with greater than 50% of microhaplotype loci missing were excluded from subsequent analyses.

Expected heterozygosity ($H_e$) at a locus was calculated using custom R code as $H_e = [\frac{n}{n-1}][1 - \sum p^2]$ (equation 1), where n is the number of samples, and p is the allele frequency of each microhaplotype allele at the locus. The variance of $H_e$ was calculated according to the formula: $2(n-1)/n^3 \{2(n-2)[\sum(p^3 - (\sum p^2)^2)]\}$ (equation 2). The within-host complexity of *P. falciparum* infections was calculated using a Markov chain Monte Carlo approach from the 100 microhaplotypes with the highest $H_e$ (R package MOIRE, https://github.com/EPPIcenter/moire). Monogenomic infections were considered when the complexity of infection was one. The geographic structure was tested using Random Forest classification[68] (R package randomForest, with ntree = 2500) on microhaplotypes with $H_e$ in the top 25 percentile as predictors and the geographical location (province and region) as the outcome. Balanced training datasets (representing 75% of the data) were used for initial testing, and test datasets (the remaining 25% of data) were used to calculate the out-of-bag error rate of the classification model. An out-of-bag error rate lower than 25% was considered as a reasonably good classification. Visualization of the classification was performed by a Principal Coordinates Analysis of the proximity matrix. Microhaplotypes in the 50 kb region flanking *pfdhps* were used to infer the evolutionary histories of mutant alleles. Population structure was visualized using t-distributed stochastic neighbor embedding, which considers the presence/absence of microhaplotype calculated with the R package Rtsne with 10000 iterations. Between sample relatedness was assessed through pairwise IBS calculated as $\frac{1}{n}\sum_{i=1}^{n} Si/XiYi$ (equation 3), where *n* is the number of loci, *Si* is the number of microhaplotype alleles shared by the samples at locus *i*, and *Xi* and *Yi* are the number of microhaplotype alleles at locus *i* of samples *X* and *Y* respectively. Chi-square test and logistic regression models were used to compare frequencies of resistance markers between regions and study periods. Differences in $H_e$ between *pfdhps* haplotypes at individual microhaplotype loci was tested through a permutation test which randomly shuffled the labels of the sub-populations 1000 times at each microhaplotype locus[69]. Kruskal–Wallis rank sum test was used for the comparison of the distribution of $H_e$ and IBS between populations, with Dunn's test and Bonferroni correction for multiple testing in pairwise comparisons.

**Statistics and reproducibility**. This study analyzed 2251 samples collected from 40 districts in seven provinces from Mozambique. Among these, sequencing produced at least one resistance-associated genotype in 1784 samples, which were included for statistical analysis. Whole genome sequences were obtained from a total of 1452 samples which passed quality filters. Statistical analyses were performed in Stata version 15.0 and R version 4.1.2. All reported *p* values are two-sided, and a *p* value of less than 0.05 was considered to indicate statistical significance.

**Reporting summary**. Further information on research design is available in the Nature Portfolio Reporting Summary linked to this article.

## Data availability

The sequences have been deposited in the European Nucleotide Archive (ENA) under Project Name PRJEB2136 and the Sequence Read Archive (SRA) under BioProject ID PRJNA910151. A deidentified and restricted dataset can be provided by approved request after completion of a data use agreement by emailing to the corresponding author. The source data for all graphs are provided in Supplementary Data 1 and Figshare (https://figshare.com/s/1920d5bad8268218b480 [Fig. 3c] and https://figshare.com/s/464a6825e09691aec654 [Fig. 3d]).

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

## Acknowledgements

Special thanks to the study participants who donated blood samples for molecular analysis, as well as clinicians and nurses who helped with the data and sample collection. This work was supported by the Bill and Melinda Gates Foundation (INV-019032 and OPP1132226), the National Institute of Health (1R01AI123050), the Departament d'Universitats i Recerca de la Generalitat de Catalunya (AGAUR; 2021SGR01517 and 2022FIB00148 fellowship for S.B.), Ministerio de Ciencia e Innovación in Spain

(PID2020-118328RB-I00), the European Union's Horizon 2020 research and innovation program under the Marie Skłodowska-Curie (grant 890477), and the US Agency for International Development through the US President's Malaria Initiative. CISM is supported by the Government of Mozambique and the Spanish Agency for International Development (AECID). We also acknowledge support from the grant CEX2018-000806-S funded by MCIN/AEI/ 10.13039/501100011033 and support from the Generalitat de Catalunya through the CERCA Program. This research is part of ISGlobal's Program on the Molecular Mechanisms of Malaria, which is partially supported by the Fundación Ramón Areces. This publication uses data from the MalariaGEN SpotMalaria project as described in 'Jacob CG et al.; Genetic surveillance in the Greater Mekong Subregion and South Asia to support malaria control and elimination; eLife 2021;10:e62997 https://doi.org/10.7554/eLife.62997. The SpotMalaria project is coordinated by the MalariaGEN Resource Centre with funding from Wellcome (206194, 090770). The authors would like to thank the staff of Wellcome Sanger Institute Sample Management, Genotyping, Sequencing, and Informatics teams, and the personnel at CISM and ISGlobal laboratory and clinic department for their contribution. Whole genome sequencing data was also produced by the Chan Zuckerberg Biohub (CZB) through BG's role as a CZB Investigator, with very helpful support from Norma Neff and the rest of the CZB genomics team. The findings and conclusions in this report are those of the author(s) and do not necessarily represent the official codon of the US Centers for Disease Control and Prevention or the US Agency for International Development. The funders had no role in the study design, data collection, data analysis, data interpretation, or writing of this manuscript. The corresponding author had full access to all the data in the study and had final responsibility for the decision to submit for publication.

## Author contributions

Wrote the first draft of the manuscript: A.M., C.d.S., and S.B. Provided technical support for the laboratory work and analyzed molecular data: P.C., A.C., N.H., B.G., S.T., and E.R. Coordinated fieldwork activities: A.N., B.G., G.M., L.N., and A.C. Performed the bioinformatic analysis: D.D. and N.H. Conceived and designed the study protocols of the original studies where samples came from: P.A., F.S., B.C., E.M., P.L.A., S.E., C.G., B.G., E.d.C., B.G., M.M.P., J.C., A.P., A.S., R.Z., and A.M. Participated in interpretation of results: A.P., E.R.-V., D.D., A.A.-D., and A.M. Provided comments to the manuscript: A.P., A.N., E.R.-V., C.G., R.Z., B.G., S.B., B.G., M.M.P., S.B., B.G., P.L.A., and A.A.-D. Reviewed and approved the final manuscript: All.

## Competing interests

The authors declare no competing interests.
