## [Peer Review File · Communications Biology]

Reviewers' comments:

Reviewer #1 (Remarks to the Author):

da Silva et al present an interesting study of population structure and resistance evolution in on of the hot-spots of malaria occurrence. The study is sound and the topic is of great importance.

The study has limitations due to its necessarily opportunistic sampling. The spatial resolutions seems relatively good, as data is presented at the level of districts, the temporal resolution with a comparison of only two time points (2015 vs. 2018) is very limited. These limitations sufficiently discussed and the authors nicely use them to derived recommendations for recording of the emergence of resistance (e.g. when treatment recommendations are changed). If it would be possible to associate resistance mutation frequencies at a smaller spatial or temporal scale with changes in drug treatment it would be easier to draw conclusions and eventually derive recommendations. Maybe this discussion could be strengthened with some deeper statistical analysis of the main narrative of the paper.

This is my only major concern: the statistical analysis of the temporal (2015 vs 2018) and North-South differences in resistance genotype should be refined.

I got this impression from multiple places in the manuscript:
abstract l42 "This resistance gradient was overlaid with... "
or discussion l 304 -305 "was also overlaid on a regional separation"

-> "overlaid with" is very vague, arguably too vague to base a main narrative of the paper on it.

Given the sophistication of statistical approaches used to e.g. distinguish mono-infection from infection with multiple genotypes and to estimate population structure, I find the approach used for the detection of the frequency differences in the occurrence of resistance mutations slightly underwhelming.

"Chi-square test was used to compare frequencies of resistance markers between provinces and study periods."

I would see two straight forward statistical analyses to address this

1. Logistic regression models (with presence or absence of a mutation as response) and temporal and spatial predictors.

and/or

2. Multivariate analysis of (e.g. Jaccard) distances (presence/absence of mutations) between samples and/or PERMANOVA or gradient fitting in nMDS to associate predictor variables with gradients in the multivariate space.

Models in both methods could be analysed for differences in mutations with vs without reported resistance phenotype or even for differences between coding and non-coding mutations (by e.g. including the type of mutation as an additional predictor).

Discussion

l 275 - 277 contradicts itself, saying that pfk13 only mutations "not associated with delayed parasite clearance" were discovered in the samples. But then one mutation that is actually still "associated with delayed parasite clearance" is mentioned. I guess this means all the mutations were

not associated with the mutation most commonly associated, but for one of them there is some indication for an association? This just needs more clear writing.

l 295 - 298 are a very nice summary of the motivation behind the diversity (IBS and He) analyses. I'd like to see a bit more of this conceptual, "goal-oriented" perspective in the results, figure legends and in the abstract (instead of the vague mutations "overlaid with" diversity). I think this would help the reader to understand the study better when reading (primarily) other parts of the manuscript.

l 285. "previous study in Mozambique which identified 1.1% which found multiple copies of plasmepsin 2 in 1.1% of" grammatical error /doubling of a half-sentence.

l 326 -327 "Second, the appropriate performance of piperaquine as an ACT partners drug". This sentence is missing a verb.

l 361 I'd add "the 50kb region around pfdhps which overlap coding regions OF OTHER GENES".

Figure 1 does not have a figure legend and I can't follow what the numbers in the table inserts mean.

Figure 3: Both E and F represent "genetic complexity estimates" for spatial (region) and are differentiated by time (2015 / 2018). Maybe use one sentence indicating that regional differences are displayed and point to labels as "E (2015) and F (2018)". Additionally, the text uses "COI" (for Complexity Of Infection) the figure legend uses "MOI" without spelling out the (alternative) acronym.

I would generally appreciate attempts to make it easier to understand the manuscript by reading each part separately: the abstract, figures results text alone. This might be a matter of taste, but I prefer more redundancy in the individual parts of a manuscript in order to "tell the the full story in each part".

Reviewer #2 (Remarks to the Author):

Overall, I have been impressed by the approach and presentation of the paper. Such studies could be important to predict and monitor possible onset of resistance towards particular antimalarials thus helping policy formulation and drug usage recommendations. I feel that studies of this nature, although having certain limitations are needed for continuous surveillance in malaria endemic regions.

Minor comment :

Please check supplementary tables 5 and 6. Maybe, the title/ legends can be further refined to enhance understanding.

Reviewer #3 (Remarks to the Author):

North-south divide in *P. falciparum* drug resistance markers 1 and genetic structure in Mozambique by da Silva et al.

In this study authors -used amplicon-based and whole genome sequencing, together with machine-learning approaches, to describe the spatial and temporal distribution of antimalarial drug resistance markers and geographic structure of *P. falciparum* parasites collected in 2015 and 2018 across south, central and north Mozambique.- All sections of the manuscript, Figures and Tables are clearly presented, and there are only minor concerns to address and listed below:

1) I suggest to move lines 175-176 to the method's section.

"A total of 2251 samples were collected in 2015 (n=724) and 2018 (n=1527) from 40 districts in seven provinces from Mozambique (Supplementary Table 1-2)."

2) I do not understand very well some numbers of paragraph (lines 185-187),

"Among the 1296 *P. falciparum* samples successfully genotyped for *pfkelch13*, 1392 were"

For 1392....It must be less than 1296? Please clarify.

3) This seems that not all markers were obtained for the 1784 samples? If it is true, a table showing all these data would be very helpful.

4) Lines 227-229: why mean H_e was calculated using 8722 microhaplotypes, weren't they reduced by excluding 349?...if not please clarify

"A total of 8722 microhaplotype loci were reconstructed via local assembly from 1438 samples which produced whole genome sequences. Of these, 349 samples contained data for less than 50 percent of all microhaplotype loci (Supplementary Fig. 3) and were therefore excluded."

5) The complexity of infection is indicated on line 246 as COI, however I suspect that correspond to MOI indicated in Figure 3? if true please be consistent, or what is MOI in Figure 3, please add that to the figure caption.

6) Please change (line 247) "differ" to "differed" as results represent the past (2015-2018).

7) Please revise "in vivo" "in vitro" should be in italics e.g. lines 289, 301, 323,

8) Discussion: It is observed that mutations in *PfKelch13* were different between 2015 and 2018 (Table 1), only one mutation was shared (likely forming two clusters), it has some relation with other mutations of the analyzed markers? or regions?.

We are very grateful to the reviewers for the time spent in revising the document and providing constructive comments. We have addressed them all and hope that the quality of the manuscript increases in this revised version.

Reviewer #1:

da Silva et al present an interesting study of population structure and resistance evolution in on of the hot-spots of malaria occurrence. The study is sound and the topic is of great importance. The study has limitations due to its necessarily opportunistic sampling. The spatial resolutions seems relatively good, as data is presented a the level of districts, the temporal resolution with a comparison of only two time points (2015 vs. 2018) is very limited. These limitations sufficiently discussed and the authors nicely use them to derived recommendations for recording of the emergence of resistance (e.g. when treatment recommendations are changed). If it would be possible to associate resistance mutation frequencies at a smaller spatial or temporal scale with changes in drug treatment it would be easier to draw conclusions and eventually derive recommendations. Maybe this discussion could be strengthened with some deeper statistical analysis of the main narrative of the paper.

Comment 1: This is my only major concern: the statistical analysis of the temporal (2015 vs 2018) and North-South differences in resistance genotype should be refined. I got this impression from multiple places in the manuscript: abstract l42 "This resistance gradient was overlaid with... " or discussion l 304 -305 "was also overlaid on a regional separation". "overlaid with" is very vague, arguably too vague to base a main narrative of the paper on it. Given the sophistication of statistical approaches used to e.g. distinguish mono-infection from infection with multiple genotypes and to estimate population structure, I find the approach used for the detection of the frequency differences in the occurrence of resistance mutations slightly underwhelming: "Chi-square test was used to compare frequencies of resistance markers between provinces and study periods." I would see two straight forward statistical analyses to address this: 1. Logistic regression models (with presence or absence of a mutation as response) and temporal and spatial predictors and/or 2. Multivariate analysis of (e.g. Jaccard) distances (presence/absence of mutations) between samples and/or PERMANOVA or gradient fitting in nMDS to associate predictor variables with gradients in the multivariate space. Models in both methods could be analysed for differences in mutations with vs without reported resistance phenotype or even for differences between coding and non-coding mutations (by e.g. including the type of mutation as an additional predictor).

Response: We have conducted, as suggested by the reviewer, a multivariable logistic regression model, with presence or absence of a mutation as response, and temporal and spatial predictors. As shown in the table below, the output of the model confirms the previous results: the relative abundance of *dhfr/dhps* mutations increases from north to south and from 2015 to 2018. Similarly, the logistic analysis showed that region, but not period, was associated with the proportion of monoclonal infections. In contrast, no statistically significant changes by region and year were observed for other resistance markers. We have included this information in the manuscript (methods, results and supplementary table).

We opted to use the term "overlay" to refer to the coincidence of several observations (increasing frequency *pf**dhfr/dhps* mutations from north to south, reduction towards south in the genetic complexity of *P. falciparum* infections and a signal of geographic differentiation), in an attempt to emphasize the lack of proven causality, as the study does not allow to demonstrate causal relationships. Even the logistic regression model does not allow us to establish such causality. We have changed the term "overlay" by "coincident", "associated" or "accompanied", to express the associations found in the analysis.

	Region			Period		
	OR	(95%CI)	p	OR	(95%CI)	p
Monoclonal (n=1090)						
North	1		0.013	2015	1	0.464
Central	1.29	(0.87; 1.90)		2018	0.88	
South	1.62	(1.16; 2.26)				
dhfr 51 (n=1638)						

	North	1			2015	1			0.009
	Central	1.37	(0.63; 2.98)	0.033	2018	2.29	(1.23; 4.29)		
	South	2.67	(1.25; 5.69)						
dhfr 59 (n=1625)									
	North	1			2015	1			0.002
	Central	3.38	(1.35; 8.45)	<0.001	2018	3.4	(1.59; 7.29)		
	South	7.66	(2.93;20.04)						
dhfr 108 (n=1649)									
	North	1			2015	1			0.004
	Central	3.1	(0.92;10.48)	0.005	2018	5.06	(1.67;15.34)		
	South	11.38	(2.39;54.25)						
dhfr pure (n=1600)									
	North	1			2015	1			0.002
	Central	1.53	(0.76; 3.06)	0.006	2018	2.46	(1.40; 4.33)		
	South	2.98	(1.51; 5.89)						
dhps 436 (n=1539)									
	North	1			2015	1			0.683
	Central	0.09	(0.04; 0.19)	<0.001	2018	0.87	(0.44; 1.71)		
	South	0.01	(0.00; 0.04)						
dhps 437 (n=1439)									
	North	1			2015	1			0.004
	Central	3.28	(2.18; 4.95)	<0.001	2018	1.78	(1.20; 2.63)		
	South	17.55	(10.73;28.72)						
dhps 540 (n=1404)									
	North	1			2015	1			0.003
	Central	2.94	(1.96; 4.41)	<0.001	2018	1.75	(1.20; 2.55)		
	South	12.92	(8.22;20.32)						
dhps pure (n=1377)									
	North	1			2015	1			0.009
	Central	3.22	(2.12; 4.89)	<0.001	2018	1.69	(1.14; 2.50)		
	South	15.03	(9.32;24.24)						
quintuple pure (n=1330)									
	North	1			2015	1			0.007
	Central	2.77	(1.85; 4.14)	<0.001	2018	1.67	(1.15; 2.42)		
	South	11.27	(7.33;17.33)						
mdr1 184 (n=1171)									
	North	1			2015	1			0.354
	Central	0.78	(0.54; 1.11)	0.291	2018	1.14	(0.87; 1.50)		
	South	0.93	(0.68; 1.29)						

Comment 2 (Discussion): I 275 - 277 contradicts itself, saying that *pfkelch13* only mutations "not associated with delayed parasite clearance" were discovered in the samples. But then one mutation that is actually still "associated with delayed parasite clearance" is mentioned. I guess this means all the mutation were not associated with the mutation most commonly associated, but for one of them there is some indication for an association? This just needs more clear writing.

Response: We thank the reviewer for pointing this contradiction. We have made the following corrections: "The *pfkelch13* wild-type, artemisinin-sensitive haplotype predominated in the parasite population surveyed from Mozambique, and none of the artemisinin-resistant validated variants were detected (ref. 6). However, an array of 32 rare non-synonymous mutations that have not been associated with delayed

parasite clearance⁶ were identified, similarly to results from other studies in Africa³⁶. Among them, the N537D mutation, which has been reported as potentially associated with delayed clearance³⁶, **was detected in one sample, whereas** A578S, the most predominant *pfkelch13* mutation in *P. falciparum* parasites of African origin³⁶ but not associated with artemisinin tolerance, **was found in four of the 1429 genotyped samples.**”

Comment 3: | 295 - 298 are a very nice summary of the motivation behind the diversity (IBS and He) analyses. I'd like to see a bit more of this conceptual, "goal-oriented" perspective in the results, figure legends and in the abstract (instead of the vague mutations "overlayed with" diversity). I think this would help the reader to understand the study better when reading (primarily) other parts of the manuscript.

Response: We have clarified the goal-oriented perspective in the manuscript by detailing the motivation of the different analysis at:

- the end of the Introduction: “In this study, we used amplicon-based and whole genome sequencing, machine-learning approaches, and relatedness as well as diversity analysis of microhaplotypes flanking *pfdhps* to describe the spatial and temporal distribution of antimalarial drug resistance markers, the geographic structure of *P. falciparum* parasites, and the evolutionary history of *pfdhps* mutant alleles in samples collected in 2015 and 2018 across south, central and north Mozambique”.
- in the Data analysis section: “The analysis aimed at describing the spatial and temporal distribution of antimalarial drug resistance markers, the geographic structure of *P. falciparum* parasites and the evolutionary history of *pfdhps* mutant alleles”.
- In the legend of Figure 3: “Microhaplotypes from regions of 150-300 bp in length between long tandem repeats were reconstructed from whole genome sequences and used to test the geographic structure of *P. falciparum* parasites”.
- In the legend of Figure 4: “Identify by state (IBS) and expected heterozygosity was calculated using 16 microhaplotypes flanking *pfdhps* to assess the evolutionary history of *pfdhps* mutant alleles in Mozambique”.

We opted not to change the abstract (due to limited space) and the results (which already contains an explanation of the motivation for the diversity analysis: “Microhaplotypes flanking *pfdhps* were used to infer the evolutionary history of the mutant alleles in Cabo Delgado, as in the rest of provinces double mutants had almost reached fixation (frequencies between 80% and 100%)”.

Comment 4: | 285. "previous study in Mozambique which identified 1.1% which found multiple copies of plasmepsin 2 in 1.1% of" grammatical error /doubling of a half-sentence.

Response: Thanks for pointing this mistake. We have made the following correction: "previous study in Mozambique ~~which identified 1.1%~~ which found multiple copies of plasmepsin 2 in 1.1% of"

Comment 5: | 326 -327 "Second, the appropriate performance of piperazine as an ACT partners drug". This sentence is missing a verb.

Response: Thanks again! We have completed the sentence, which was missing some words: “**The lack of mutations at codon 415 of *pfexo* and the low prevalence of plasmepsin2/3 breakpoints detected (0.4%) suggests** the appropriate performance of piperazine as an ACT partners drug”.

Comment 6: | 361 I'd add "the 50kb region around *pfdhps* which overlap coding regions OF OTHER GENES”.

Response: Added “of other genes” as suggested.

Comment 7: Figure 1 does not have a figure legend and I can't follow what the numbers in the table inserts mean.

Response: We have detailed the legend of Figure 1: "Source of *P. falciparum* samples providing genetic data. **Tables indicate the number of samples included in the analysis per province and year, for each of the three main regions of the country. Provincial borders are indicated with thick lines. The specific districts providing data for the study are colored**".

Comment 8: Figure 3: Both E and F represent "genetic complexity estimates" for spatial (region) and are differentiated by time (2015 / 2018). Maybe use one sentence indicating that regional differences are displayed and point to labels as "E (2015) and F (2018)". Additionally, the text uses "COI" (for Complexity Of Infection) the figure legend uses "MOI" without spelling out the (alternative) acronym.

Response: We have corrected the legend as suggested to clarify the regional differences: "**E) and F) Genetic complexity estimates Complexity of infection (COI) for samples in different regions of Mozambique in 2015 (E) and 2018 (F), as indicated by the number of genetically distinct clones**". We have also changed MOI to COI in the figure.

Comment 9: I would generally appreciate attempts to make it easier to understand the manuscript by reading each part separately: the abstract, figures results text alone. This might be a matter of taste, but I prefer more redundancy in the individual parts of a manuscript in order to "tell the full story in each part".

Response: As described above (comment 3), we have detailed motivations of the different analysis in each section (Introduction, Methods, Results and Figure legends) to facilitate the understanding of the manuscript.

Reviewer #2

Overall, I have been impressed by the approach and presentation of the paper. Such studies could be important to predict and monitor possible onset of resistance towards particular antimalarials thus helping policy formulation and drug usage recommendations. I feel that studies of this nature, although having certain limitations are needed for continuous surveillance in malaria endemic regions.

Comment 1 (Minor comment): Please check supplementary tables 5 and 6. Maybe, the title/ legends can be further refined to enhance understanding.

Response: We have followed reviewer's suggestion and refined the titles and legends for an easier understanding:

Supplementary Table 5. Frequency of *P. falciparum* isolates carrying mutations in *pf dhfr*, *pf dhps* and *pfmdr1* molecular markers of antimalarial resistance by province and year. Samples with a mixed genotype in any of the positions *pf dhfr*-51, -59 or -108, and *pf dhps*-437 or -540 were excluded. ***pf dhfr* and *pf dhps* haplotypes were built as a combination of mutations at these positions. *pf dhfr* double mutants were defined as any combination of two mutations out of the three possible. *pf dhps* single mutants were defined as the presence of either *pf dhps*-437 or -540.** P values indicate the statistical significance of the difference in frequencies between provinces.

Supplementary Table 6. Frequency of *P. falciparum* isolates carrying mutations in *pf dhfr*, *pf dhps* and *pfmdr1* by year.

Samples with a mixed genotype in any of the positions ***pf dhfr*-51, -59 or -108, and *pf dhps*-437 or -540 were excluded. *pf dhfr* and *pf dhps* haplotypes were built as a combination of mutations at these positions. *pf dhfr* double mutants were defined as any combination of two mutations out of the three possible. *pf dhps* single mutants were defined as the presence of either *pf dhps*-437 or -540.** P values indicates the statistical significance of the difference in frequencies between years.

Reviewer #3

In this study authors -used amplicon-based and whole genome sequencing, together with machine-learning approaches, to describe the spatial and temporal distribution of antimalarial drug resistance markers and geographic structure of *P. falciparum* parasites collected in 2015 and 2018 across south, central and north Mozambique.- All sections of the manuscript, Figures and Tables are clearly presented, and there are only minor concerns to address and listed below:

Comment 1: I suggest to move lines 175-176 to the method's section: "A total of 2251 samples were collected in 2015 (n=724) and 2018 (n=1527) from 40 districts in seven provinces from Mozambique (Supplementary Table 1-2)."

Response: We have followed the suggestion of the reviewer and moved the sentence to methods section.

Comment 2: I do not understand very well some numbers of paragraph (lines 185-187), "Among the 1296 *P. falciparum* samples successfully genotyped for *pfkelch13*, 1392 were". For 1392....It must be less than 1296? Please clarify.

Response: We are very grateful to the reviewer for identifying this mistake in the numbers. We have made the following correction: "Among the **1429** *P. falciparum* samples successfully genotyped for *pfkelch13*, **1393** were fully wild-type and **36 (2.5%)** presented a total of 32 non-synonymous mutations".

Comment 3: This seems that not all markers were obtained for the 1784 samples? If it is true, a table showing all these data would be very helpful.

Response: The reviewer is correct: not all the markers were obtained for the 1784 samples. To make this clear, we have included a supplementary table showing the number of samples for which markers were obtained:

	2015	2018	Total
kelch13	285	1144	1429
pfdhfr			
51	399	1268	1667
59	390	1253	1643
108	396	1261	1657
164	362	1209	1571
pfdhps			
436	367	1251	1618
437	364	1252	1616
540	348	1252	1581
581	310	1180	1490
613	312	1178	1490
pfmdr1			
86	387	1218	1605
184	363	1173	1536
1246	361	1158	1519
pfcr1			
72	390	1265	1655
74	389	1268	1657
75	390	1268	1658
76	389	1267	1656
Artemisinin-resistance genetic background			
127	402	1304	1706

	128	401	806	1207
	193	383	1231	1614
	326	395	1247	1642
	356	393	1247	1640
	484	364	1201	1565
pfexo				
	415	394	1000	1394
Plasmepsin 2/3 breakpoint				0
		108	416	524

Comment 4: Lines 227-229: why mean He was calculated using 8722 microhaplotypes, weren't they reduced by excluding 349?...if not please clarify: "A total of 8722 microhaplotype loci were reconstructed via local assembly from 1438 samples which produced whole genome sequences. Of these, 349 samples contained data for less than 50 percent of all microhaplotype loci (Supplementary Fig. 3) and were therefore excluded."

Response: We excluded 349 samples (due to low-quality sequencing results) from a total of 1438 samples which were sequenced. However, the number of microhaplotypes we analyzed was 8722, and this number did not change. So, we calculated the mean He for 8722 using data from 1089 (1438-349) samples. We have clarified this in the manuscript: "The median He of the 8722 microhaplotypes **from 1089 samples with data for more than 50% of the microhaplotypes** was 0.312"

Comment 5: The complexity of infection is indicated on line 246 as COI, however I suspect that correspond to MOI indicated in Figure 3? if true please be consistent, or what is MOI in Figure 3, please add that to the figure caption.

Response: We have changed MOI to COI in figure 3.

Comment 6: Please change (line 247) "differ" to differed" as results represent the past (2015-2018).

Response: Thanks, corrected.

Comment 7: Please revise "in vivo" "in vitro" should be in italics e.g. lines 289, 301, 323,

Response: Thanks again, corrected.

Comment 8: Discussion: It is observed that mutations in PfKelch13 were different between 2015 and 2018 (Table 1) , only one mutation was shared (likely forming two clusters), it has some relation with other mutations of the analyzed markers? or regions?

Response: As suggested by the reviewer, we have analyzed more in detailed the 4 samples which presented a kelch13 mutation (A578S) observed in 2015 and 2018 (see table below). However, we have not observed any relationship with other resistance mutations nor regions.

District	Province	Region	Year	pfcr1	pfdhfr	pfdhps	pfexo	pfmdr1	art-R gb
Moatize	Tete	Central	2015	CVMNK	IRNI	SGEAA	E	NXD	VDDNIT
Montepuez	C. Delgado	North	2018	CVMNK	IRNI	SXXAA	E	NXD	VDDNIT
Montepuez	C. Delgado	North	2018	CVMNK	IRNI	SGXAA	E	NFD	VDDNIT
Moatize	Tete	Central	2018	CVMNK	IRNI	SGEAA	E	NFD	VDDNIT

gb, genetic background

Other changes:

- We have corrected a mistake in the ENA accession numbers (from PRJEB126 to PRJEB2136), in the number of samples analyzed for some specific markers and other minor mistakes detected during the revision that do not affect the content of the manuscript (all in track changes).
- We have included supplementary methods to main manuscript.
- We have corrected a mistake in results section (Sixteen microhaplotypes instead of seventeen):
“**Sixteen** microhaplotypes were contained in a 50kb region around the gene pfdhps, **15** of them in 9 genes and one intergenic”
- We have completed the legend of figure 4 with the sample size and statistical test used.
- We have added to Supplementary materials the data underlying the main figures in the manuscript:
 - o Supplementary Data 1. Complexity of infection (COI) obtained from *P. falciparum* samples included in the study (data used for Figure 3E and F).
 - o Supplementary Data 2. Presence or absence in samples included in the study of the 16 microhaplotypes contained in a 50kb region around the gene pfdhps (data from Figure 4B).
 - o Supplementary Data 3. Expected heterozygosity calculated from the 16 microhaplotype loci in a 50 kb region around pfdhps in parasites collected from Cabo Delgado (data from Figure 4c)

REVIEWERS' COMMENTS:

Reviewer #1 (Remarks to the Author):

The authors updated their manuscript and provide an additional statistical test, which was the main point of my review. I very much appreciate the clear layout and information reporting the results of this test in supplementary table 8. The association of most, but not all, specific mutations with the temporal and spatial contrasts is strong. This might be interesting for other readers too. I appreciate the changes in wording from "overlaid" to "coincident" or "associated", as I find the latter more standard terminology when pointing out statistical difference (overlaid sound to me more like an approach to comparison, while the updated wording points more to a result).

The authors also have made small but very efficient changes in the writing to convey the message of the manuscript more clearly. In my opinion this improved the cohesion between results and discussion. The manuscript now reads very well throughout.

I have no further comments and complement the authors on their study and its presentation.

Reviewer #3 (Remarks to the Author):

North-south divide in *P. falciparum* drug resistance markers and genetic structure in Mozambique by da Silva et al.

In my opinion, the authors have responded to most reviewer's concerns accordingly. However, there are still some minor points to revise

Lines 161-164: add to numbers "codons" e.g. "pfkelch13 (any mutation in codons 349-726..." to any section of the text, tables or figures, or supplementary materials, if necessary.

Please revise lines 255-258, if parenthesis or square brackets apply to be consistent across the text "codon 51", "codon 437" instead, etc. revise this and other paragraphs "in position 51 [1596/1638], 98% in 59 [1597/1625] and 99% in 108 [1635/1649]) and $\geq 88\%$ in pfdhps (90% in 437 [1289/1439] and 88% in 540 [1242/1404]; Supplementary Table 6 & Supplementary Fig 2). The most prevalent pfdhfr and pfdhps alleles were the triple (S108N/N51I/C59R; 99% [1548/1600]) and double mutants (A437G/K540E; 89% [1228/1377]), respectively, with an 87% (1155/1330) of quintuple "

Line 270: Is it correct? "at position 436 (S436CAHF) in pfdhps"

I did not find the following correction in the manuscript, please indicate lines: "The pfkelch13 wild-type, artemisinin-sensitive haplotype predominated in the parasite population surveyed from Mozambique, and none of the artemisinin-resistant validated variants were detected (ref. 6). However, an array of 32 rare non-synonymous mutations that have not been associated with delayed parasite clearance⁶ were identified, similarly to results from other studies in Africa³⁶. Among them, the N537D mutation, which has been reported as potentially associated with delayed clearance³⁶, was detected in one sample, whereas A578S, the most predominant pfkelch13 mutation in *P. falciparum* parasites of African origin³⁶ but not associated with artemisinin tolerance, was found in four of the 1429 genotyped samples.

Revise the text of the manuscript, in tables or figures, genus and species are in italics. E.g. Figure 3,

Check spaces: e.g. line 192,

We are very grateful to the reviewers for the time spent in revising the document and providing constructive comments. We have addressed them all and hope that the quality of the manuscript increases in this revised version.

Reviewer #3:

In my opinion, the authors have responded to most reviewer's concerns accordingly. However, there are still some minor points to revise

Lines 161-164: add to numbers "codons" e.g. "pfkelch13 (any mutation in codons 349-726..." to any section of the text, tables or figures, or supplementary materials, if necessary.

Response: Corrected as suggested.

Please revise lines 255-258, if parenthesis or square brackets apply to be consistent across the text "codon 51", "codon 437" instead, etc. revise this and other paragraphs "in position 51 [1596/1638], 98% in 59 [1597/1625] and 99% in 108 [1635/1649]) and ≥88% in pfdhps (90% in 437 [1289/1439] and 88% in 540 [1242/1404]; Supplementary Table 6 & Supplementary Fig 2). The most prevalent pfdhfr and pfdhps alleles were the triple (S108N/N511/C59R; 99% [1548/1600]) and double mutants (A437G/K540E; 89% [1228/1377]), respectively, with an 87% (1155/1330) of quintuple "

Response: Revised and corrected.

Line 270: Is it correct? "at position 436 (S436CAHF) in pfdhps"

Response: Revised and corrected.

I did not find the following correction in the manuscript, please indicate lines: "The pfkelch13 wild-type, artemisinin-sensitive haplotype predominated in the parasite population surveyed from Mozambique, and none of the artemisinin-resistant validated variants were detected (ref. 6). However, an array of 32 rare non-synonymous mutations that have not been associated with delayed parasite clearance⁶ were identified, similarly to results from other studies in Africa³⁶. Among them, the N537D mutation, which has been reported as potentially associated with delayed clearance³⁶, was detected in one sample, whereas A578S, the most predominant pfkelch13 mutation in *P. falciparum* parasites of African origin³⁶ but not associated with artemisinin tolerance, was found in four of the 1429 genotyped samples.

Response: This correction is present in 216-220.

The original version was:

"The *pfkelch13* wild-type, artemisinin-sensitive haplotype predominated in the parasite population surveyed from Mozambique. However, an array of 32 rare non-synonymous mutations that have not been associated with delayed parasite clearance⁶ were identified, similarly to results from other studies in Africa³⁶. Among them, the N537D mutation, which has been reported as potentially associated with delayed clearance³⁶, and A578S, the most predominant *pfkelch13* mutation in *P. falciparum* parasites of African origin³⁶ but not associated with artemisinin tolerance, were carried by one and four samples, respectively, out of the 1296 genotyped samples".

Following reviewer's suggestion, we reworded it to:

"The *pfkelch13* wild-type, artemisinin-sensitive haplotype predominated in the parasite population surveyed from Mozambique, and none of the artemisinin-resistant validated variants were detected⁶. However, an array of 32 rare non-synonymous mutations were identified, similarly to results from other studies in Africa³⁷. Among them, the N537D mutation, which has been reported as potentially associated with delayed clearance³⁷, was detected in one sample, whereas A578S, the most predominant *pfkelch13* mutation in *P.*

falciparum parasites of African origin³⁷ but not associated with artemisinin tolerance, was found in four of the 1429 genotyped samples”.

Revise the text of the manuscript, in tables or figures, genus and species are in italics. E.g. Figure 3,

Response: Corrected.

Check spaces: e.g. line 192,

Response: Corrected.